# Online Test-Time Adaptation in Tabular Data
# with Minimal High-Certainty Samples

Zhiqing Xiao * [1]   Mingming Zhang * [1]   Junbo Zhao [1]

## Abstract

Tabular data is ubiquitous across real-world applications, yet most representation learning methods still assume an unrealistic IID setting. In practice, tabular streams often exhibit mixed covariate and label shifts, making existing domain generalization or vision-oriented test-time adaptation methods ineffective. We propose a simple yet effective **O**nline **T**est-**T**ime **A**daptation approach for **T**abular data (OT3A). OT3A uses high-confidence and domain-consistent pseudo-labels to estimate and correct target label distribution shifts, then applies self-training and entropy minimization on these reliable samples for online adaptation. Extensive experiments across diverse distribution shift scenarios show that OT3A significantly outperforms existing methods, highlighting its efficacy and practicality for real-world tabular adaptation.

## 1. Introduction

Tabular data is widely utilized across real-world application scenarios (Zhou et al., 2018; Guo et al., 2021; Chen et al., 2016; Sadar et al., 2023; Abdou & Pointon, 2011), and deep neural networks have recently shown strong performance for tabular representation learning (Chen et al., 2022; Yoon et al., 2020; Popov et al., 2020; Huang et al., 2020; Somepalli et al., 2021). However, these models are usually developed under an independent and identically distributed (IID) assumption. In practice, deployed tabular data often changes across environments, so the test distribution can differ substantially from the training distribution and cause severe performance degradation.

Test-Time Adaptation (TTA) offers a way to adapt a deployed model to unlabeled target data, and has been widely studied in computer vision (Sun et al., 2020b; Wang et al., 2021; Niu et al., 2022), natural language processing (Hardt & Sun, 2024; Banerjee et al., 2021), and speech processing (Bai et al., 2023; Huang et al., 2024). Yet TTA for tabular data remains largely underexplored, and directly applying methods from other domains often yields suboptimal results because tabular data lack the augmentation and structural priors commonly used in vision. We identify two key challenges: tabular data are frequently imbalanced, which biases predictions toward the source-domain distribution, and their shifts are often mixed, involving covariate shift, label shift, and concept drift rather than a single clean assumption.

Real-world tabular streams also require online adaptation, where the model processes incoming target data in mini-batches and updates sequentially without revisiting source data. We therefore focus on Online Test-Time Adaptation (OTTA) (Azizzadenesheli et al., 2019; Park et al., 2023; Alexandari et al., 2020). Standard OTTA tools such as pseudo-label learning (Lee, 2013) and entropy minimization (Niu et al., 2022; 2023) can be effective, but they are sensitive to noisy target predictions and may reinforce errors under imbalance. This makes reliable sample selection essential: highly confident and locally consistent samples can provide stronger guidance for adapting to the target distribution.

In this work, we propose **O**nline **T**est-**T**ime **T**abular **A**daptation, termed OT3A. OT3A leverages high-confidence and domain-consistent pseudo-labels to estimate and correct target label distribution shifts, and then uses self-training and entropy minimization on these reliable samples to adapt online to out-of-distribution test data. This design allows the model to update with each arriving target mini-batch while reducing the effect of noisy pseudo-labels.

Our main contributions are as follows:

- We investigate the OTTA problem for tabular data with detailed analysis to highlight its core challenges: The co-existence of label and covariate distribution shifts, and even class imbalance.

- We propose a novel and practical method, OT3A, which leverages high-certainty samples to estimate the

*Equal contribution  [1]College of Computer Science and Technology, Zhejiang University, Hangzhou, China. Correspondence to: Mingming Zhang <mmz@zju.edu,cn>.

*Proceedings of the 2^{nd} ICML Workshop on Foundation Models for Structured Data*, Seoul, South Korea. 2026. Copyright 2026 by the author(s).

label distribution and integrates entropy minimization with pseudo-label learning to simultaneously address both label and covariate shifts.

- We demonstrate through extensive experiments that OT3A achieves significant performance improvements over existing methods under various distribution shift settings.

## 2. Background

We study source-free online test-time adaptation, where an initial model $f_{\theta_0}$ is trained on labeled source data $\mathcal{D}_S = \{(x_i^S, y_i^S)\}_{i=1}^{N_S}$ and then deployed on an unlabeled target stream. At each time step $t$, the model receives only the current target mini-batch $\mathcal{D}_T^t = \{x_i^{T,t}\}_{i=1}^{N_t}$. We consider a closed-set setting with shared input and label spaces, and the target distribution differs from the source distribution. The goal is to adapt the model using only the current unlabeled batch and predict its target labels without revisiting source data or observing target labels.

This setting is particularly important for tabular data, which are common in finance, healthcare, recommendation, and other streaming applications where heterogeneous features, class imbalance, and distribution shift are prominent (Zhou et al., 2018; Guo et al., 2021; Chen et al., 2016; Abdou & Pointon, 2011). At the same time, many test-time adaptation methods from vision do not transfer well to tabular tasks because tabular data lack the augmentation and structural priors of image-like modalities (Azizzadenesheli et al., 2019; Alexandari et al., 2020). Although recent work has begun to explore tabular test-time adaptation (Ren et al., 2024; Fang et al., 2024; Kim et al., 2024; Zhou et al., 2025), existing methods often depend on specialized model structures, extra training stages, output-only calibration, or assumptions centered on a limited shift type. These limitations motivate a unified online framework that can jointly handle mixed shift, class imbalance, and noisy pseudo-labels.

## 3. Online Framework with Sample Selection and Distribution Calibration

### 3.1. Overall Framework

OT3A follows a three-stage online pipeline: reliable sample selection, target-distribution estimation, and online calibration. For each target mini-batch, we first construct a target dynamic graph and jointly select reliable samples using neighborhood consistency and prediction confidence. We then estimate the target label distribution by combining model predictions with graph-based propagation over the selected samples, which reduces the instability of relying on a single estimation source. Finally, we use the estimated distribution to recalibrate model outputs and update the model

with pseudo-label supervision and entropy minimization. This design aims to suppress error accumulation from noisy pseudo-labels while adapting to imbalanced, and mixed-shift target streams.

### 3.2. Sample Selection

Under online test-time adaptation, updating on all predictions in the current batch is vulnerable to pseudo-label noise. We therefore construct a target dynamic graph and extract reliable target samples for later distribution estimation and online optimization.

**Target dynamic graph construction.** At time step $t$, given the unlabeled target mini-batch $\mathcal{D}_T^t = \{x_i^{T,t}\}_{i=1}^{N_t}$, we define the neighborhood of each sample $x_i^{T,t}$ by a feature-similarity measure $d(\cdot, \cdot)$:

$$\mathcal{N}(x_i^{T,t}) = \left\{ x_j^{T,t} \in \mathcal{D}_T^t \;\middle|\; d(x_i^{T,t}, x_j^{T,t}) \leq h_t \right\}, \quad (1)$$

where $h_t$ is the neighborhood radius at time step $t$. This defines a target dynamic graph with a row-normalized adjacency matrix $\mathbf{A}_t \in \mathbb{R}^{N_t \times N_t}$.

The model prediction for sample $x_i^{T,t}$ is $\mathbf{p}_i^t = \mathrm{softmax}(f(x_i^{T,t}; \theta_{t-1})) \in [0,1]^{C_T}$, where $\theta_{t-1}$ denotes the current model parameters, and its pseudo-label is $\hat{y}_i^t = \arg\max_{c \in \mathcal{Y}_T}[\mathbf{p}_i^t]_c$.

**Reliable sample selection.** High-confidence samples generally provide more reliable predictions, but using a single global threshold can make majority classes dominate the selected set. We therefore adopt class-wise adaptive quantile thresholds and define the high-confidence sample set as

$$\mathcal{S}_{\mathrm{cf}}^t = \left\{ x_i^{T,t} \in \mathcal{D}_T^t \;\middle|\; [\mathbf{p}_i^t]_{\hat{y}_i^t} \geq \tau_{\hat{y}_i^t} \right\}, \quad (2)$$

where $\tau_c$ is the confidence quantile threshold for samples predicted as class $c$ in the current batch.

Following Kim et al. (2024), confidence alone is not enough to suppress confirmation bias, so we further measure neighborhood consistency by $s_i^t = |\mathcal{N}(x_i^{T,t})|^{-1} \sum_{x_j^{T,t} \in \mathcal{N}(x_i^{T,t})} \mathbb{I}(\hat{y}_i^t = \hat{y}_j^t)$. Samples above the consistency quantile threshold $\tau_s$ form $\mathcal{S}_{\mathrm{cs}}^t$, and the final reliable sample set is $\mathcal{S}_{\mathrm{sel}}^t = \mathcal{S}_{\mathrm{cf}}^t \cap \mathcal{S}_{\mathrm{cs}}^t$, which provides a more stable basis for the subsequent stages.

### 3.3. Distribution Estimation

**Prediction-based class-distribution estimation.** Using the soft predictions $\mathbf{p}_i^t \in [0,1]^{C_T}$, we estimate the target class distribution as $\hat{\boldsymbol{\pi}}_t^{\mathrm{pred}} = \frac{1}{N_t} \sum_{i=1}^{N_t} \mathbf{p}_i^t \in \mathbb{R}^{C_T}$. This estimate reflects the overall prediction tendency of the current batch, but it may inherit source-prior bias under label shift.

**Target-graph-based class-distribution estimation.**
Based on the reliable sample set $\mathcal{S}_{\text{sel}}^t$, we form an initial label matrix $\mathbf{Q}_t \in \mathbb{R}^{N_t \times C_T}$, whose $i$-th row is the one-hot pseudo-label of $x_i^{T,t}$ if $x_i^{T,t} \in \mathcal{S}_{\text{sel}}^t$, and $\mathbf{0}$ otherwise.

We then inject $\mathbf{Q}_t$ as label anchors into the target dynamic graph and propagate labels through the neighborhood:

$$\mathbf{F}_t^{(\ell+1)} = \alpha \mathbf{A}_t \mathbf{F}_t^{(\ell)} + (1-\alpha)\mathbf{Q}_t,$$

where $\alpha \in (0,1)$ is the propagation coefficient and $\mathbf{F}_t^{(0)} = \mathbf{0}$. After convergence, we obtain a soft-label matrix $\mathbf{F}_t^* \in \mathbb{R}^{N_t \times C_T}$ and estimate each class proportion by

$$\hat{\pi}_{t,c}^{\text{graph}} = \frac{1}{N_t} \sum_{i=1}^{N_t} \mathbb{I}\left(\arg\max_j [\mathbf{F}_t^*]_{i,j} = c\right).$$

This estimate exploits target-domain local structure and is often more stable under class imbalance.

**Fusion of class-distribution estimates.** The two estimates above emphasize complementary information from model predictions and target-domain local structure. To adaptively balance them, we compute an imbalance coefficient $\zeta_t$ as the $\ell_2$ distance between the empirical class histogram induced by $\mathbf{Q}_t$ on $\mathcal{S}_{\text{sel}}^t$ and the uniform distribution:

$$\zeta_t = \left[ \sum_{c=1}^{C_T} \left( \frac{1}{|\mathcal{S}_{\text{sel}}^t|} \sum_{x_i^{T,t} \in \mathcal{S}_{\text{sel}}^t} [\mathbf{Q}_t]_{i,c} - \frac{1}{C_T} \right)^2 \right]^{1/2}.$$

The fused target class-distribution estimate is

$$\hat{\boldsymbol{\pi}}_t = (1-\zeta_t)\hat{\boldsymbol{\pi}}_t^{\text{pred}} + \zeta_t \hat{\boldsymbol{\pi}}_t^{\text{graph}}. \tag{3}$$

When the mini-batch becomes more imbalanced, $\zeta_t$ increases and the method relies more heavily on the graph-based estimate; otherwise it retains more of the soft prediction prior.

### 3.4. Online Calibration

**Distribution calibration.** If the source-domain class prior $\boldsymbol{\pi}_S \in \mathbb{R}^{C_T}$ differs from the current target distribution, the model may produce biased predictions. Because the true target prior is unavailable at test time, we approximate it with $\hat{\boldsymbol{\pi}}_t$ from §3.3 and recalibrate each posterior by $z_{i,c}^t = p_{i,c}^t \hat{\pi}_{t,c} / \pi_{S,c}$ and $\tilde{p}_{i,c}^t = z_{i,c}^t / \sum_{j=1}^{C_T} z_{i,j}^t$. The resulting $\tilde{\mathbf{p}}_i^t$ corrects prediction bias induced by label shift.

**Online update.** After distribution calibration, the model further adapts online to mitigate the decision-boundary mismatch caused by covariate shift. Rather than updating on all target samples, we optimize only over the reliable sample set $\mathcal{S}_{\text{sel}}^t$. Using $\tilde{y}_i^t = \arg\max_{c \in \mathcal{Y}_T} [\tilde{\mathbf{p}}_i^t]_c$, we define the pseudo-label supervision loss $\mathcal{L}_{\text{pl}}^t = \mathcal{L}_{\text{ce}}(\tilde{\mathbf{p}}_i^t, \tilde{y}_i^t)$ and the

entropy-minimization term $\mathcal{L}_{\text{ent}}^t = -\sum_{c=1}^{C_T} [\tilde{\mathbf{p}}_i^t]_c \log[\tilde{\mathbf{p}}_i^t]_c$. The online optimization objective at time step $t$ is

$$\mathcal{L}_{total} = \frac{1}{|\mathcal{S}_{\text{sel}}^t|} \sum_{x_i^{T,t} \in \mathcal{S}_{\text{sel}}^t} \left( \mathcal{L}_{\text{pl}}^t + \lambda \mathcal{L}_{\text{ent}}^t \right), \tag{4}$$

where $\lambda$ is a trade-off coefficient. In this way, the model continuously adapts to the target distribution without access to source data or target labels.

## 4. Experimental Results

### 4.1. Experimental Setup

We evaluate OT3A on five representative TableShift datasets using two deep tabular backbones, MLP (Rumelhart et al., 1986) and TabTransformer (Huang et al., 2020); more experimental details and additional results are provided in Appendix A.

### 4.2. Main Experimental Analysis

Table 1 reports Acc, bAcc, and F1 under MLP and Tab-Transformer backbones; because Acc can be dominated by majority classes under mixed shifts, we focus on bAcc and F1. Diabetes provides a representative example: under both backbones, several baselines retain high Acc but clearly weaker bAcc and F1. Under MLP, PL reaches 83.81% Acc but only 55.10% bAcc and 55.30% F1, whereas OT3A improves them to 71.29% and 67.87%; under TabTransformer, the source model attains 83.23% Acc but only 54.42% bAcc and 54.08% F1, while OT3A reaches 73.62% bAcc and 67.67% F1. This confirms that Acc alone can be misleading and that the proposed distribution calibration effectively alleviates severe class bias.

The gains of OT3A are also consistent across datasets and backbones. With MLP, the source model reaches only 53.25% bAcc on HELOC, while OT3A raises it to 64.12%, and on Diabetes the bAcc improves from 55.22% to 71.29%; similar improvements also appear on ASSISTments and Hospital Readmission. With TabTransformer, OT3A still delivers clear gains on datasets with pronounced label shift, improving HELOC from 59.31% to 62.67% in bAcc and from 50.12% to 62.67% in F1, and raising Voting from 73.87% to 79.00% in bAcc and from 74.74% to 79.12% in F1. Overall, these results show that OT3A remains effective across realistic shift scenarios rather than only under a single favorable condition.

### 4.3. Model Analysis

**Ablation analysis.** Table 2 evaluates the effectiveness of OT3A's key components on HELOC, Voting, Diabetes, AS-SISTments, and Hospital Readmission. The first part ablates the sample-selection module, and the second part ablates the online-calibration module. Overall, the complete

*Table 1.* Overall performance analysis under the MLP and TabTransformer backbones

| Backbone | Method | HELOC | | | Voting | | | Diabetes | | | ASSISTments | | | Hospital Readmission | | |
|---|---|---|---|---|---|---|---|---|---|---|---|---|---|---|---|---|
| | | Acc | bAcc | F1 | Acc | bAcc | F1 | Acc | bAcc | F1 | Acc | bAcc | F1 | Acc | bAcc | F1 |
| MLP | Source Only | 53.21 ±4.3 | 53.25 ±3.5 | 40.02 ±5.3 | 78.69 ±0.3 | 75.66 ±0.4 | 77.24 ±0.2 | 83.32 ±0.2 | 55.22 ±0.1 | 55.50 ±0.0 | 51.57 ±3.2 | 60.81 ±3.3 | 46.42 ±1.8 | 60.65 ±0.3 | 60.59 ±0.2 | 59.12 ±1.6 |
| | PL (Lee, 2013) | 54.78 ±1.1 | 51.82 ±1.3 | 34.92 ±2.3 | 75.87 ±0.4 | 75.61 ±0.3 | 76.63 ±0.5 | **83.81** ±0.1 | 55.10 ±0.1 | 55.30 ±0.1 | 56.45 ±1.3 | 57.30 ±0.6 | 44.49 ±0.5 | 60.27 ±0.2 | 60.65 ±0.0 | 59.16 ±0.6 |
| | TTT (Sun et al., 2020a) | 53.05 ±3.1 | 53.20 ±1.5 | 38.21 ±3.6 | 76.08 ±0.5 | 76.80 ±0.5 | 77.64 ±0.3 | 83.29 ±0.0 | 55.41 ±0.2 | 55.73 ±0.1 | 55.86 ±1.3 | 60.02 ±0.2 | 45.92 ±0.3 | 61.02 ±0.3 | 60.46 ±0.0 | 59.62 ±0.1 |
| | TENT (Wang et al., 2021) | 54.35 ±2.3 | 54.24 ±5.8 | 39.91 ±2.6 | 78.07 ±0.6 | 74.09 ±0.6 | 74.76 ±0.3 | 83.32 ±0.0 | 55.00 ±0.0 | 55.00 ±0.0 | 50.87 ±0.3 | 56.41 ±0.3 | **63.99** ±0.2 | 61.34 ±0.3 | 60.15 ±0.3 | 53.75 ±1.0 |
| | EATA (Niu et al., 2022) | 54.37 ±2.1 | 54.25 ±3.6 | 40.02 ±1.6 | 78.13 ±0.3 | 76.20 ±0.5 | 77.79 ±0.1 | 83.50 ±0.3 | 55.20 ±0.4 | 55.50 ±0.1 | 55.86 ±0.2 | 60.81 ±0.2 | 46.42 ±0.1 | 61.36 ±0.3 | 60.16 ±0.3 | 53.68 ±0.9 |
| | LAME (Boudiaf et al., 2022) | 43.10 ±4.6 | 50.00 ±0.0 | 30.10 ±0.6 | 63.50 ±2.1 | 54.60 ±0.4 | 46.80 ±0.1 | 83.24 ±0.1 | 54.82 ±0.2 | 54.80 ±0.6 | 45.12 ±0.1 | 51.30 ±0.1 | 41.40 ±0.1 | **61.39** ±0.1 | 54.90 ±0.3 | 46.69 ±1.4 |
| | ODS (Zhou et al., 2023) | 43.10 ±4.6 | 50.00 ±0.0 | 30.10 ±0.6 | 63.50 ±2.1 | 54.60 ±0.5 | 46.80 ±0.1 | 83.24 ±0.1 | 54.80 ±0.3 | 54.80 ±0.2 | 45.12 ±0.1 | 51.30 ±0.1 | 41.40 ±0.1 | **61.39** ±0.1 | 54.90 ±0.3 | 46.69 ±1.4 |
| | SAR (Niu et al., 2023) | 52.32 ±2.6 | 54.74 ±0.9 | 33.16 ±2.9 | 78.13 ±0.6 | 64.20 ±0.5 | 59.79 ±0.1 | 82.98 ±0.2 | 53.48 ±0.4 | 54.81 ±0.9 | 55.86 ±0.2 | 60.81 ±0.1 | 46.42 ±0.1 | 61.38 ±0.2 | 57.19 ±0.3 | 51.98 ±0.9 |
| | OT3A | **64.56** ±1.9 | **64.12** ±0.9 | **63.97** ±0.7 | **80.21** ±0.4 | **78.32** ±0.5 | **78.97** ±0.2 | 79.91 ±0.2 | **71.29** ±0.5 | **67.67** ±0.2 | **62.28** ±0.4 | 63.97 ±1.1 | 61.89 ±0.6 | 61.03 ±0.2 | **61.03** ±0.2 | 59.92 ±0.9 |
| Tab Transformer | Source Only | 55.13 ±1.5 | 59.31 ±0.5 | 50.12 ±0.3 | 77.89 ±0.1 | 73.87 ±0.0 | 74.74 ±0.1 | **83.23** ±0.1 | 54.42 ±0.1 | 54.08 ±0.0 | 45.22 ±3.2 | 51.34 ±3.3 | 33.72 ±1.8 | 61.49 ±0.3 | 61.03 ±0.2 | 60.44 ±1.6 |
| | PL (Lee, 2013) | 54.26 ±0.1 | 58.72 ±0.3 | 51.44 ±0.3 | 77.25 ±0.4 | 72.97 ±0.3 | 73.53 ±0.5 | 83.00 ±0.3 | 52.39 ±0.1 | 50.32 ±0.2 | 44.12 ±0.4 | 50.41 ±0.3 | 31.32 ±0.5 | 60.29 ±0.2 | 60.00 ±0.0 | 57.83 ±0.6 |
| | TTT (Sun et al., 2020a) | 54.15 ±0.5 | 58.21 ±0.4 | 49.31 ±0.6 | 77.18 ±0.5 | 74.10 ±0.5 | 73.64 ±0.2 | 83.10 ±0.0 | 52.13 ±0.2 | 52.41 ±0.1 | 44.16 ±1.3 | 50.02 ±0.6 | 35.92 ±0.2 | 60.46 ±0.3 | 59.46 ±0.0 | 59.62 ±0.1 |
| | TENT (Wang et al., 2021) | 54.35 ±2.3 | 57.24 ±5.8 | 49.93 ±2.6 | 78.07 ±0.6 | 74.05 ±0.6 | 74.16 ±0.3 | 83.12 ±0.0 | 52.00 ±0.0 | 52.20 ±0.0 | 40.87 ±0.3 | 46.41 ±0.3 | 33.99 ±0.2 | 61.12 ±0.3 | 60.15 ±0.3 | 53.75 ±1.0 |
| | EATA (Niu et al., 2022) | 55.07 ±0.1 | 57.25 ±0.2 | 50.02 ±0.6 | 78.03 ±0.1 | 76.20 ±0.5 | 74.79 ±0.1 | 83.00 ±0.1 | 52.12 ±0.2 | 52.05 ±0.1 | 45.86 ±0.3 | 50.81 ±0.3 | 36.42 ±0.1 | 61.36 ±0.3 | 60.16 ±0.3 | 53.68 ±0.9 |
| | LAME (Boudiaf et al., 2022) | 49.15 ±0.6 | 56.05 ±0.0 | 50.10 ±0.6 | 73.50 ±0.1 | 74.60 ±0.4 | 74.80 ±0.1 | 82.54 ±0.2 | 52.83 ±0.2 | 52.56 ±0.6 | 45.12 ±0.2 | 48.79 ±0.3 | 31.40 ±0.1 | 60.05 ±0.1 | 58.90 ±0.3 | 46.69 ±1.4 |
| | ODS (Zhou et al., 2023) | 54.10 ±0.6 | 57.00 ±0.0 | 51.14 ±0.2 | 77.50 ±2.1 | 74.60 ±0.5 | 74.80 ±0.0 | 82.26 ±0.1 | 54.16 ±0.1 | 52.06 ±0.1 | 45.12 ±0.2 | 51.30 ±0.1 | 32.08 ±0.1 | 61.39 ±0.1 | 60.12 ±0.3 | 59.25 ±1.4 |
| | SAR (Niu et al., 2023) | 54.32 ±2.6 | 56.74 ±0.9 | 48.18 ±2.9 | 77.13 ±0.6 | 74.20 ±0.5 | 73.79 ±0.1 | 82.98 ±0.2 | 53.48 ±0.4 | 52.81 ±0.9 | 45.86 ±0.2 | 50.81 ±0.1 | 36.42 ±0.1 | 60.38 ±0.2 | 57.89 ±0.3 | 58.48 ±0.9 |
| | OT3A | **63.39** ±0.7 | **62.67** ±0.6 | **62.67** ±0.5 | **79.96** ±0.1 | **79.00** ±0.1 | **79.12** ±0.2 | 76.52 ±0.2 | **73.62** ±0.1 | **67.67** ±0.2 | **56.93** ±0.4 | **53.94** ±1.1 | **52.55** ±0.6 | **61.70** ±0.2 | **61.71** ±0.2 | **61.70** ±0.9 |

*Table 2.* Ablation analysis

| Variant | HELOC | | Voting | | Diabetes | | ASSISTments | | Hospital Readmission | |
|---|---|---|---|---|---|---|---|---|---|---|
| | bAcc | F1 | bAcc | F1 | bAcc | F1 | bAcc | F1 | bAcc | F1 |
| OT3A ($\mathcal{S}_{sel}^t + \mathcal{L}_{total}^t$) | **64.12** ±0.9 | **63.97** ±0.7 | **78.32** ±0.5 | **78.97** ±0.2 | **71.29** ±0.5 | **67.87** ±0.2 | **63.97** ±1.1 | **61.89** ±0.6 | **61.03** ±0.2 | **59.92** ±0.9 |
| w/o $\mathcal{S}_{sel}^t$ ($\mathcal{S}_{cf}^t \cap \mathcal{S}_{cs}^t$) | 57.78 ±0.3 | 50.03 ±0.1 | 74.87 ±0.2 | 75.84 ±0.2 | 56.57 ±0.1 | 57.55 ±0.2 | 63.04 ±0.5 | 54.53 ±0.5 | 60.41 ±0.4 | 59.10 ±0.2 |
| w/o $\mathcal{S}_{cf}^t$ | 63.64 ±0.5 | 62.35 ±0.6 | 76.16 ±0.5 | 77.14 ±0.3 | 66.25 ±0.1 | 64.21 ±0.1 | 59.54 ±0.6 | 48.87 ±0.4 | 57.21 ±0.1 | 51.65 ±0.3 |
| w/o $\mathcal{L}_{total}^t$ ($\mathcal{L}_{ent}^t + \mathcal{L}_{pl}^t$) | 64.07 ±0.3 | **63.97** ±0.3 | 77.87 ±0.4 | 78.59 ±0.1 | 69.71 ±0.0 | 67.82 ±0.1 | 62.65 ±0.1 | 55.94 ±0.3 | 56.21 ±0.7 | 49.37 ±0.2 |
| w/o $\mathcal{L}_{ent}^t$ | 63.97 ±0.4 | 63.86 ±0.3 | 77.88 ±0.5 | 78.59 ±0.1 | 69.97 ±0.1 | 67.82 ±0.1 | 63.20 ±0.3 | 61.71 ±0.3 | 56.62 ±0.3 | 50.26 ±0.1 |
| Source Only | 53.25 ±3.5 | 40.02 ±5.3 | 75.66 ±0.4 | 77.24 ±0.2 | 55.22 ±0.1 | 55.50 ±0.0 | 60.81 ±3.3 | 46.42 ±1.8 | 60.59 ±0.2 | 59.12 ±1.6 |

OT3A pipeline achieves the best bAcc and F1 across the five datasets, which confirms the necessity of the core modules.

More specifically, removing the sample-selection module, i.e., w/o $\mathcal{S}_{sel}^t$, leads to the most obvious performance degradation, especially on HELOC, Diabetes, and ASSISTments. This shows that selecting reliable target samples is crucial for stable target-information estimation and for suppressing the influence of noisy samples. Further, removing only the high-confidence component, i.e., w/o $\mathcal{S}_{cf}^t$, still causes substantial degradation on several datasets, especially on AS-SISTments and Hospital Readmission, where the F1 score drops to 48.87% and 51.65%, respectively. This demonstrates that the high-confidence subset is an essential source of trustworthy pseudo-supervision.

Similarly, removing the online-calibration module also weakens performance. Both w/o $\mathcal{L}_{total}^t$ and w/o $\mathcal{L}_{ent}^t$ produce clear degradation on multiple datasets, with especially large F1 drops on ASSISTments and Hospital Readmission. This indicates that the optimization losses defined in the online-calibration stage effectively improve target-domain discrimination and help balance precision and recall.

## 5. Conclusion

This paper studies online test-time adaptation for tabular data under online streams. We propose OT3A, an online adaptation framework based on sample selection and distribution-shift calibration, to alleviate performance degradation caused by mixed distribution shift. The method consists of three key components: reliable sample selection, target-distribution estimation, and online calibration. It uses a target dynamic graph to identify reliable target samples, estimate the target label distribution, and continuously adapt the model without access to source data or target priors. Experimental results show that OT3A achieves stable and significant improvements on multiple real tabular datasets, especially in scenarios with class imbalance and distribution change, highlighting its robustness and practical potential.

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

# A. Additional Experimental Details

This appendix provides the experimental information omitted from the main text due to space constraints, including dataset descriptions, implementation details, additional motivation, and supplementary analyses.

## A.1. Dataset Summary

*Table 3.* Dataset statistics for the main experiments

| Statistic | HELOC | Voting | Diabetes | ASSISTments | Hospital Readmission |
|---|---|---|---|---|---|
| Total instances | 9,412 | 60,376 | 1,299,758 | 2,400,998 | 89,542 |
| Train | 2,220 | 34,796 | 969,229 | 2,132,526 | 34,288 |
| Validation | 278 | 4,349 | 121,154 | 266,566 | 4,286 |
| Test | 6,914 | 21,231 | 209,375 | 1,906 | 50,968 |
| Total features | 22 | 54 | 25 | 26 | 46 |
| Numerical features | 20 | 8 | 6 | 9 | 12 |
| Categorical features | 2 | 46 | 19 | 17 | 34 |

**Dataset descriptions.** Following recent studies on online test-time adaptation for tabular data, we evaluate OT3A on five representative TableShift datasets with naturally occurring distribution shifts. HELOC predicts whether an applicant will repay a home equity line of credit within two years; the shift is induced by external risk assessment, where lower-risk-score samples are assigned to the test split. Hospital Readmission contains clinical records of diabetic patients from U.S. hospitals and healthcare networks from 1999 to 2008, and the task is to predict whether a patient will be readmitted within 30 days; distribution shift is introduced through differences in admission sources. Voting is derived from the American National Election Studies and predicts U.S. presidential-election voting behavior, with domain shift induced by geographic regions. Diabetes contains approximately 1.4 million observations for diabetes prediction, where the shift mainly reflects race and ethnicity differences in diabetes risk. ASSISTments comes from an online tutoring platform and involves predicting student affective or behavioral states, such as boredom, confusion, frustration, and engaged problem solving.

## A.2. Experimental Setup Details

**Backbones.** We use two representative deep tabular backbones. MLP (Rumelhart et al., 1986) is a standard fully connected neural network and serves as a simple but strong baseline architecture for tabular prediction. TabTransformer (Huang et al., 2020) uses self-attention to model interactions among categorical and numerical features, providing a stronger feature-interaction modeling architecture than a plain MLP.

**Baselines.** We compare OT3A with representative test-time adaptation baselines. PL (Lee, 2013) updates the network by minimizing cross-entropy between predictions and pseudo-labels. TTT (Sun et al., 2020a) introduces an auxiliary self-supervised objective to mitigate test-time distribution shift. TENT (Wang et al., 2021) updates model parameters by minimizing prediction entropy on target data. EATA (Niu et al., 2022) combines active sample selection with Fisher regularization to improve stability and reduce forgetting. LAME (Boudiaf et al., 2022) adapts output probabilities with Laplacian regularization without explicit parameter updates. ODS (Zhou et al., 2023) models mixed distribution shifts by separately considering covariate and label shifts. SAR (Niu et al., 2023) filters samples according to prediction entropy and encourages updates toward flat minima for robust adaptation.

**Metrics and protocol.** All methods are evaluated in the source-free online test-time adaptation setting: the source model is trained only before deployment, source data are inaccessible during adaptation, and target test samples are

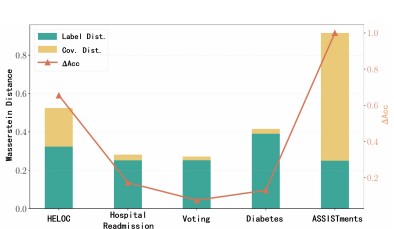

*(a)* Mixed covariate and label shifts.

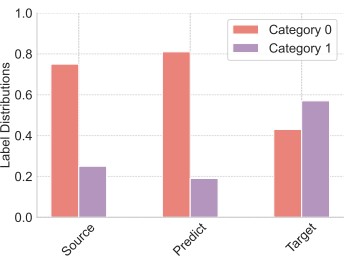

*(b)* Prediction bias under imbalance.

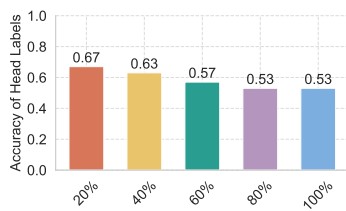

*(c)* High-confidence samples are more reliable.

*Figure 1.* Three empirical observations motivating OT3A. The panels show that tabular streams contain mixed shifts, source-prior bias, and a small set of reliable high-confidence samples.

processed sequentially in mini-batches. We report accuracy (Acc), balanced accuracy (bAcc), and macro F1. Since tabular target streams are often class-imbalanced, Acc can be dominated by majority classes, so bAcc and F1 are the primary indicators. All reported results are means and standard errors over three independent runs.

**Implementation details.** We follow Gardner et al. (2023) for source-model training. For both backbones, source models are trained with mini-batch size 512, AdamW optimizer, learning rate 0.01, and weight decay 0.01; the number of training epochs is selected according to validation performance. During test-time adaptation, OT3A uses a confidence quantile threshold of 0.25, a consistency quantile threshold of 0.7, a propagation coefficient of 0.8, and a Gaussian-kernel graph parameter of $\sigma = 1$, unless otherwise specified.

**Baseline hyperparameters.** For PL, TENT, and SAR, the main adaptation hyperparameters are learning rate, number of update steps per target mini-batch, and whether episodic adaptation is used, i.e., whether model parameters are reset after each test batch. PL and TENT use learning rate $1 \times 10^{-4}$, one adaptation step per batch, and episodic adaptation. SAR uses learning rate $1 \times 10^{-3}$, one adaptation step per batch, episodic adaptation, and its entropy-based filtering threshold. For TTT, we use a VIME-like self-supervised proxy task (Yoon et al., 2020): 15% of feature values are randomly replaced by values from the same feature column, and an auxiliary MLP is trained to identify and recover the corrupted entries. For TTT and EATA, the learning rate is $1 \times 10^{-5}$, with 10 adaptation steps per batch and episodic adaptation. Other hyperparameters follow the original papers. LAME and ODS only adjust output probabilities and therefore do not require gradient-based adaptation hyperparameters such as learning rate or update steps.

### A.3. Empirical Motivation

We highlight three empirical observations that directly motivate OT3A: realistic tabular streams usually contain mixed shifts, source-domain imbalance induces persistent prediction bias, and only a small subset of target samples is reliable enough to support safe online updates. These observations are not isolated. In online adaptation, mixed shifts increase the complexity of the target environment, label-distribution changes amplify prediction bias, and reliable high-confidence samples provide the limited but useful supervision signal available from unlabeled target data. Together, they motivate the three components of OT3A: reliable sample selection, target-distribution estimation, and online calibration.

**The coexistence of covariate and label shifts.** Figure 1a measures covariate and label shifts with Wasserstein distance across the five datasets. Different tasks often exhibit both types of shift, but their relative magnitudes vary substantially. For example, HELOC and ASSISTments show more pronounced covariate shift, whereas Voting and Diabetes are more strongly affected by label-distribution shift. This indicates that real tabular target streams rarely satisfy a single clean shift assumption; instead, the model must handle feature-distribution changes and class-prior changes simultaneously.

**Label-distribution imbalance.** Figure 1b uses HELOC to compare the source label distribution, model prediction distribution, and target label distribution. Although the true target prior differs substantially from the source prior, the model predictions remain close to the source distribution, producing a systematic mismatch. During online pseudo-label adaptation, this initial bias can be repeatedly reinforced: majority-class samples dominate the updates, while minority-class samples are more likely to be misclassified or ignored. This observation motivates explicit target-distribution estimation and calibration.

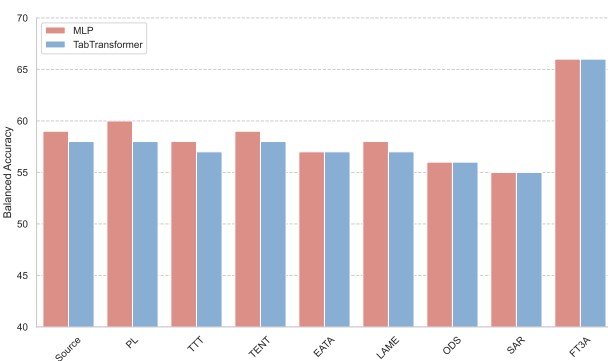

*Figure 2.* OT3A: average performance analysis (%). The results summarize performance across datasets and backbones to compare architecture-level robustness.

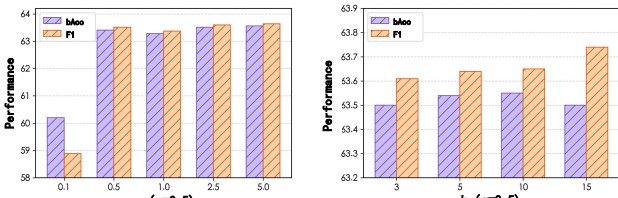

*Figure 3.* OT3A: structure analysis. The figure evaluates how graph construction parameters affect target dynamic graph robustness.

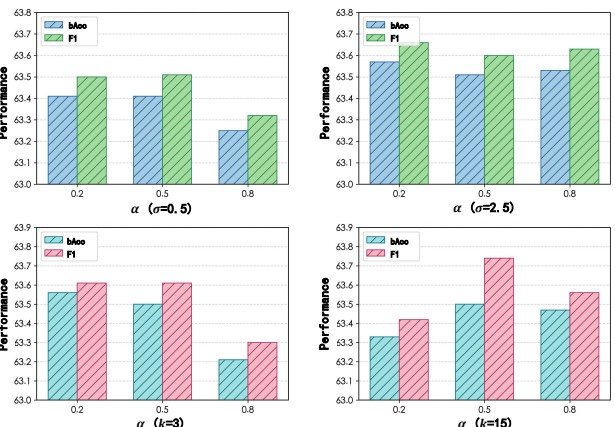

*Figure 4.* OT3A: target dynamic graph propagation analysis. The figure studies how the propagation coefficient balances graph information and noise.

**Impact of high-confidence samples.** Figure 1c analyzes prediction reliability under different confidence levels. For class 1, the top 20% high-confidence samples reach an accuracy of about $0.67$, whereas using all samples yields an accuracy of only about $0.53$. As lower-confidence samples are gradually included, prediction quality decreases, indicating that target batches contain both useful adaptation signals and noisy pseudo-labels. In particular, high-confidence minority-class samples can still preserve valuable discriminative information. The key challenge is therefore not whether useful target information exists, but how to identify reliable samples online and suppress the propagation of erroneous pseudo-labels.

## A.4. Average Performance Analysis

To further evaluate generalization at a holistic level, Figure 2 summarizes the average performance of all methods across five datasets under the two backbone models. OT3A achieves strong average performance under both backbones, and the gap between MLP and TabTransformer is small. This suggests that the advantage of OT3A does not come from a specific network architecture or model capacity, but from its shift-aware estimation and calibration mechanism itself. In contrast, other methods show limited average improvement across backbones, and some of them fluctuate noticeably when the backbone changes, indicating stronger dependence on the underlying architecture. Although TabTransformer has stronger feature-interaction modeling ability than MLP, it does not systematically outperform MLP for all adaptation methods. Under test-time distribution shift, simply increasing model capacity is therefore insufficient; explicitly modeling target-distribution changes and calibrating predictions provides more stable gains across architectures.

## A.5. Target Dynamic Graph Structure Analysis

Figure 3 analyzes the structural robustness of the target dynamic graph on HELOC with the MLP backbone. In implementation, we consider both a Gaussian-kernel graph with parameter $\sigma$ and a $k$-nearest-neighbor graph. For the Gaussian-kernel graph, when $\sigma$ increases from a very small value such as $0.1$ to a moderate range such as $0.5$ to $5.0$, performance improves substantially and then stabilizes. This indicates that overly small $\sigma$ makes the similarity distribution too sharp, producing an excessively sparse graph and weakening label propagation. Within a reasonable range, however, performance is not sensitive to the exact value of $\sigma$, demonstrating good structural robustness. For the $k$-nearest-neighbor graph, performance remains stable and slightly improves as $k$ increases, which shows that moderately enlarging the neighborhood can strengthen connectivity and improve information propagation. When $k$ varies from 3 to 15, the performance differences remain small, suggesting that the graph construction is also robust to the neighbor number. Overall, both Gaussian-kernel and $k$-nearest-neighbor graphs provide stable target dynamic graphs for OT3A.

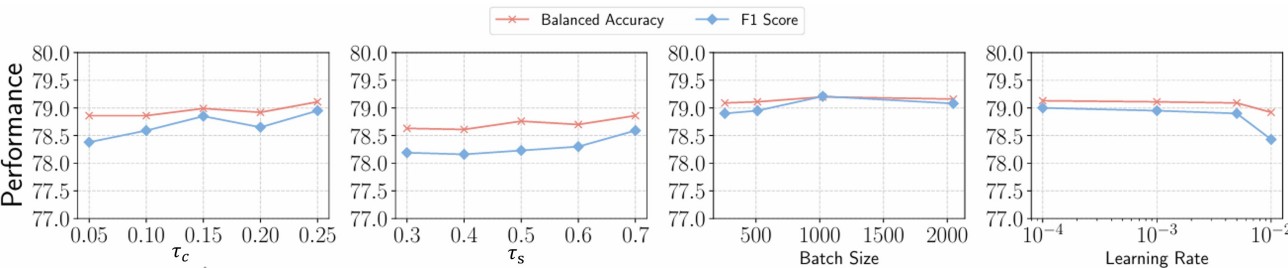

*Figure 5.* OT3A: sensitivity analysis of sample-selection hyperparameters and model parameters. The figure jointly reports the effects of selection thresholds, batch size, and learning rate.

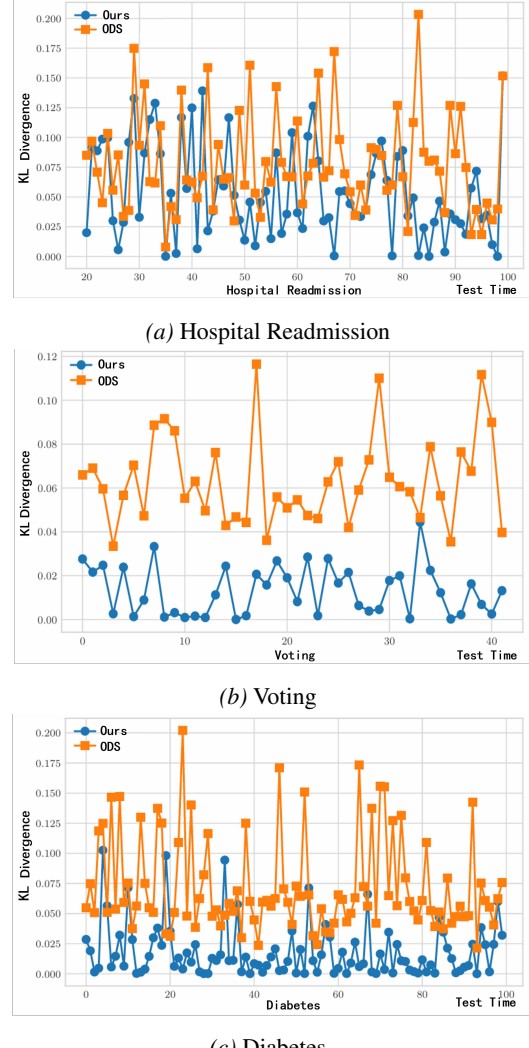

*(a)* Hospital Readmission

*(b)* Voting

*(c)* Diabetes

*Figure 6.* OT3A: distribution-estimation stability analysis. The panels compare KL divergence over time to assess how stably each method estimates target label distributions.

### A.6. Target Dynamic Graph Propagation Analysis

Figure 4 studies the effect of the propagation coefficient $\alpha$ under different graph structures. Under Gaussian-kernel graphs, increasing $\alpha$ from $0.2$ to $0.5$ slightly improves performance, while further increasing it to $0.8$ degrades performance. A similar trend appears for $k$-nearest-neighbor graphs. This shows that choosing a suitable propagation coefficient is important for balancing original predictions and graph-structural information. A small value such as $\alpha = 0.2$ may underuse neighborhood information, resulting in insufficient propagation, whereas a large value such as $\alpha = 0.8$ can make the estimate overly dependent on graph propagation and introduce neighborhood noise. The degradation is more evident when the graph is highly connected, such as when $k$ is large. Overall, moderate values such as $\alpha = 0.5$ provide the best trade-off between information propagation and noise control, supporting the stability of the proposed graph-based distribution estimation.

### A.7. Hyperparameter Sensitivity Analysis

This section studies the sensitivity of OT3A to its sample-selection hyperparameters on Voting with the MLP backbone. As shown in Figure 5, when the confidence quantile threshold varies from $0.05$ to $0.25$, both bAcc and F1 fluctuate only mildly, with a slight upward trend overall. This indicates that the high-confidence selection mechanism can stably retain useful samples across a reasonably wide threshold range. For the neighborhood-consistency quantile threshold, performance remains stable and slightly improves as the threshold increases, suggesting that stronger neighborhood-consistency constraints can improve the reliability of selected samples. When the consistency threshold varies from $0.3$ to $0.7$, the performance changes remain small, further confirming that OT3A is not overly sensitive to this threshold.

### A.8. Model-Parameter Analysis

Figure 5 also analyzes model-related parameters using Voting with the MLP backbone. Batch size has a relatively limited effect: bAcc and F1 remain stable across the tested values, with only a slight improvement at moderate batch sizes. This suggests that OT3A can estimate the target distribution and update the model reliably under different

mini-batch granularities. Smaller batches may provide less stable distribution statistics, while overly large batches can reduce the update frequency and limit the flexibility of online adaptation. By contrast, the learning rate has a more visible effect. As the learning rate increases, performance first remains stable and then improves, indicating that moderately larger steps help the model respond more quickly to target-domain drift. A learning rate that is too small may make the update insufficient to capture distribution changes, whereas an excessively large learning rate, although not harmful in the tested range, may introduce instability in more complex scenarios. Overall, OT3A requires tuning only a small number of key parameters to achieve reliable performance, which reduces practical tuning cost.

### A.9. Distribution-Estimation Stability Analysis

Figure 6 shows the change in target-label-distribution estimation error over time on Hospital Readmission, Voting, and Diabetes, comparing ODS (Zhou et al., 2023) and OT3A during adaptation. KL divergence between the estimated and true label distributions of each test batch is used as the metric. Smaller KL divergence indicates more accurate label-distribution estimation and thus more reliable distribution-aware calibration.

Overall, OT3A is consistently better than ODS on all three datasets. Its KL divergence remains lower and smoother throughout the test stream, whereas ODS exhibits larger peaks on multiple batches, indicating that its label-distribution estimate is more sensitive to prediction noise and local distribution change.

More specifically, on Hospital Readmission (Figure 6a), ODS shows large fluctuations and clear peaks at several test steps, while OT3A maintains a lower and smoother KL divergence, demonstrating more stable distribution tracking. On Voting (Figure 6b), the overall error of ODS is not always high, but its curve fluctuates more frequently; OT3A achieves a lower average error and a smoother temporal trajectory. On Diabetes (Figure 6c), the difference is more pronounced because the dataset contains stronger class imbalance and more complex distribution changes. ODS produces larger estimation errors on multiple target batches, whereas OT3A remains consistently lower and more stable.

These results validate OT3A from the perspective of label-distribution estimation. Compared with ODS, which mainly relies on model prediction statistics for distribution estimation, OT3A combines reliable sample selection with graph-based distribution estimation and distribution calibration. This reduces noise and bias in target-label-distribution estimation and provides a more reliable basis for subsequent prediction calibration and online model updates.

