# OpenReview forum: "Online Test-Time Adaptation in Tabular Data with Minimal High-Certainty Samples"
_ICML.cc/2026/Workshop/FMSD — FMSD @ ICML 2026 Poster_

### Official Review · Reviewer_WuaG · 2026-05-12
**I recommend accept and list out some areas for improvement**

**Rating:** 7
**Confidence:** 3

**Review:**

# Summary

The authors present a methodology for test-time adaptation by recalibrating its predictions based on estimates of the marginal and input-conditional label distribution derived from the model's outputs on an unlabeled minibatch of test datapoints. They report significant performance improvements over previously-proposed TTA methods for different settings.

# Strengths

- Addresses important problem -- distribution shift is huge issue in real world and often ignored in standard benchmark datasets. Setting seems realistic and useful.
- They report significant performance improvement over the considered baselines.
- Experiments seem pretty thorough -- they compare to 8 baseline methods on 5 datasets, have error bars, and have extra ablation studies and hyperparameter sensitivity analysis.
- I'm not very familiar with the related work, but from spot checking compared to the cited TabLog paper it seems like their datasets and reported performance of baselines are comparable. So on this basis their results seem plausible.

# Areas for improvement

- Needs discussion of how hyperparameters were chosen for their method and the baselines. i.e. are you looking at target domain performance when deciding which hyperparameters to use?
- Should compare to other recent tabular test-time adaptation methods, or explain why they aren't comparable.
- Method is quite complicated and some design decisions aren't clearly motivated. I would like to see clear explanations of what each component aims to achieve and why it achieves it.
- The paper considers many natural settings, but it would also be good to have 1 or 2 clean synthetic settings where you can highlight robustness to different types of distribution shift without confounding factors. And/or validate that the components of your method are doing what you claim.

# Detailed comments

- The paper should discuss their hyperparameter tuning procedure in more detail. In the distribution shift literature, a notorious problem is that methods can overfit to the target distribution if they tune hyperparameters using target domain performance, and it's unclear whether the authors have done this. I recommend referring to \[1\], which discusses this issue and provides recommendations (it's about domain generalization but still seems relevant here).
	- Examples of fair hyperparameter tuning procedures would be 1) sampling the valset from the source domain, 2) for nonstationary data you could do something like sample the trainset from time $t \in [0, 0.5]$, valset from $t \in [0.5, 0.75]$, testset from $t \in [0.75, 1]$, 3) for a discrete number of domains you could partition them into train, val, test domains and make sure test domain performance is not used for model selection.
	- If you have used target-domain performance to choose hyperparameters, this is an important limitation which should be highlighted, and I would recommend fixing it before submitting to a full conference.
- Would be nice to include a few more baselines, e.g.
	- 1 or 2 classical methods like XGBoost, logistic regression as a reference point
	- Some of the papers listed in the background section which have "begun to explore tabular test-time adaptation". Or at least clearly state why they aren't comparable/applicable.
- The method is quite complicated and all these different design decisions don't seem clearly motivated, e.g.
	- what "feature similarity measure" $d$ are we using and why? what are we trying to accomplish by computing $F_t^*,$ and why does this recursive algorithm get us there?
	- I see there are ablation studies, but it's not very clear from the row labels what is being ablated. It would be clearer to do something similar to \[2\], where you start with a vanilla model without test-time adaptation, then add training on pseudo-labels, then train on only the 'reliable' pseudo-labels, then add in calibration, then add in the entropy minimization term ...etc. and show how the performance changes as each of these different components of your method are added.
	- For each of these design decisions it would be nice to cite something, state your goal mathematically and show that the proposed method accomplishes it, or at least intuitively explain what you want to achieve and why the design decision achieves it.
- I'm skeptical of the framing that "directly applying methods from other domains often yields suboptimal results because tabular data lack the augmentation and structural priors commonly used in vision" -- AFAIK both the proposed method and most prior work don't explicitly require vision priors or data augmentation, and can be used for any classification task. It's not clear to me why the proposed method is different in this respect. IMO this needs more justification.
- Would be good to include an operating point-free performance metric such as area under ROC curve -- in my experience these are less sensitive to certain types of distribution shift.
- A few presentation changes would be nice:
	- Make clear in tables that OT3A is your method -- e.g. OT3A (ours) or \textbf{OT3A}
	- Make font, fontsize consistent for figures and use vector graphics. Ideally I would specify these + the appropriate figure width in matplotlib and avoid resizing in Latex.
	- For probability mass vectors e.g. $p_i^t$, the standard notation would be to say $p_i^t \in \Delta^{C_t - 1}$ (the set of probability distributions over $C_t$ classes) rather than $p_i^t \in [0, 1]^{C_t},$ which is imprecise.

\[1\] Gulrajani, Ishaan, and David Lopez-Paz. "In search of lost domain generalization." _arXiv preprint arXiv:2007.01434_ (2020).

\[2\] Liu, Zhuang, et al. "A convnet for the 2020s." _Proceedings of the IEEE/CVF conference on computer vision and pattern recognition_. 2022.

# Justification of score

- I think the paper merits acceptance because it addresses an underexplored important practical setting, reports strong results compared to baselines, and appears methodologically sound. Thus, it clearly meets the criteria for the workshop.
- I don't want to give a higher score because the method appears complicated/*ad hoc* without much justification for most design decisions, likely limiting usefulness/generality of the findings. I also think hyperparameter tuning methodology and comparison to prior work are somewhat lacking (or at least need more discussion).

---

### Official Review · Reviewer_qMYi · 2026-05-18
**method is well justified and supported by the experiments, and it's good to consider adding significance test as well**

**Rating:** 8
**Confidence:** 4

**Review:**

The paper proposes OT3A, an online adaptation framework that deals domain shift. The framework consists of three components, which 1. constructs a dynamic graph to select reliable samples, 2. estimate class distribution via the fusion of model prediction and distribution induced from the dynamic graph, with a self-adaptive fusion coefficient balancing the strength of model prediction and graph estimation, 3. an online calibration of the fused distribution and an online update loss. The proposed method is well-justified and the experimental results show good improvements over previous methods, however, the reviewer also suggests the authors considering adding significance test and report p-values between OT3A and the best/second-best performing method.

---

### Official Review · Reviewer_qQkZ · 2026-05-22
**Well-motivated and solid tabular TTA work, but with limited empirical evaluation**

**Rating:** 7
**Confidence:** 3

**Review:**

# Summary

This paper aims to solve the existing problem of online test-time adaptation for tabular data under realistic mixed distribution shifts and class imbalance. The authors propose OT3A, which is a framework that selects reliable high-confidence samples, estimates target label distributions using graph-based propagation, and performs online calibration with pseudo-label learning and entropy minimization. Experimental results on multiple datasets and two backbones show improvements over existing TTA baselines, especially in balanced accuracy and macro F1.

# Strengths

1. The problem framing is clear, where TTA for tabular data is a genuine research gap, and the authors present it well.
2. The proposed framework is well-motivated and practical, with reliable sample selection, distribution estimation, and online calibration directly targeting the identified challenges.
3. The paper appropriately utilizes imbalance-sensitive metrics such as balanced accuracy and macro F1 instead of completely relying on accuracy.
4. The paper is well-written.

# Areas for Improvement

1. The paper does not analyze the computational overhead of the proposed method. OT3A consists of online graph construction and graph-based label propagation during adaptation, which can add extra runtime and memory costs.
2. The baseline comparisons are not fully convincing, as comparisons against more recent tabular-specific adaptation methods are limited. In particular, the neighborhood-consistency mechanism appears closely related to prior work such as Kim et al. (2024), but this method is not included as an experimental baseline, making it difficult to clearly assess the empirical gains.
3. The connection to the workshop scope foundation models for structured data is relatively weak. The evaluated backbones are standard tabular architectures, including MLP and TabTransformer, and the paper does not discuss how the proposed method can interact with structured-data foundation models.

# Detailed Comments

1. Add runtime, latency, and scalability measurements for the graph construction and label-propagation steps across multiple batch sizes and dataset scales.
2. Include stronger tabular-TTA baselines.
3. Make the paper more related to foundation models.

# Justification of Score

OT3A addresses a real and underexplored problem with a coherent method and establish empirical results across multiple datasets and backbones, but it lacks comprehensive evaluation results and analysis, which could be improved but are not fundamental issues. Overall, I lean towards an accept for this paper.